# The Incidence of Pulmonary Hypertension and the Association with Bronchopulmonary Dysplasia in Preterm Infants of Extremely Low Gestational Age: Single Centre Study at the Maternity Hospital of University Medical Centre Ljubljana, Slovenia

**DOI:** 10.3390/children12111441

**Published:** 2025-10-24

**Authors:** Tomaž Križnar, Štefan Grosek, Tina Perme

**Affiliations:** 1Neonatal Intensive Care Unit, Neonatology Section, Division of Gynecology, Department of Perinatology, University Medical Centre Ljubljana, 1000 Ljubljana, Slovenia; stefan.grosek@kclj.si (Š.G.); tina.perme@kclj.si (T.P.); 2Department of Medical Ethics, Faculty of Medicine, University of Ljubljana, 1000 Ljubljana, Slovenia; 3Department of Paediatrics, Faculty of Medicine, University of Ljubljana, 1000 Ljubljana, Slovenia

**Keywords:** pulmonary hypertension (PH), bronchopulmonary dysplasia (BPD), BPD-associated pulmonary hypertension (BPD-PH), extremely low gestational age newborns (ELGANs), echocardiographic screening

## Abstract

**Highlights:**

**What are the main findings?**
•In the first few days of life, preterm infants with a high clinical probability of pulmonary hypertension (PH) may not yet show echocardiographic signs of PH.•In preterm infants with bronchopulmonary dysplasia (BPD), late screening prior to NICU discharge (≤36 weeks PMA) may still be too early to detect PH that develops later in the course of BPD.

**What is the implication of the main finding?**
•When there is a clear clinical indication, the absence of echocardiographic signs of PH should not be a contraindication to starting inhaled nitric oxide (iNO) or a reason to delay treatment.•Echocardiographic screening for PH should be considered beyond the neonatal period and after NICU discharge.

**Abstract:**

**Background**: Pulmonary hypertension (PH) occurs in ~25% of infants with moderate-to-severe bronchopulmonary dysplasia (BPD) and is associated with substantial morbidity and mortality. The American Heart Association and American Thoracic Society recommend routine echocardiographic screening for PH in preterm infants with BPD at 36 weeks’ postmenstrual age (PMA), yet the true incidence remains unclear owing to non-uniform diagnostic criteria. Emerging evidence suggests a potential role for earlier screening. **Objectives**: (i) to determine the incidence of pulmonary hypertension (PH) and bronchopulmonary dysplasia (BPD) in preterm infants of extremely low gestational age; (ii) to determine the incidence of PH among infants diagnosed with BPD (BPD-PH); and (iii) to evaluate the utility of early screening at 7 days of life and late screening at discharge in relation to subsequent BPD. **Methods**: We conducted a prospective cohort study of all infants born at 22 + 0 to 28 + 6 weeks’ gestation and admitted to our tertiary NICU between 1 September 2022 and 31 December 2024. Clinical and echocardiographic assessments for PH and BPD were performed by neonatologists trained in neonatal echocardiography. **Results**: Seventy-eight infants born at 22 + 0–28 + 6 weeks’ gestation were enrolled; 71 underwent early screening and 57 underwent late screening. Early echocardiography at day 7 and late screening at discharge identified no cases of PH. PH was diagnosed clinically and/or echocardiographically in 10 infants before day 7 and in one infant at 38 weeks’ PMA. BPD developed in 42 of 57 infants (73.7%). **Conclusions**: In this cohort of extremely low-gestational-age infants, echocardiographic screening performed by neonatologists detected no PH at day 7 and only one case at late screening (at 38 weeks’ PMA/before discharge). Most PH was identified prior to day 7 on clinical and/or echocardiographic grounds.

## 1. Introduction

Bronchopulmonary dysplasia (BPD) is one of the major complications of prematurity and represents the most common form of chronic lung disease in extremely low gestational age newborns (ELGANs) [1]. Pulmonary hypertension (PH) develops in approximately one-fifth of these infants, predominantly in those diagnosed with moderate-to-severe BPD [2]. The concomitant PH contributes substantially to morbidity and mortality [3]. Despite advances in pharmacotherapy, long-term outcomes for children with severe pulmonary arterial hypertension (PAH) remain poor [4].

### 1.1. Definition of Pulmonary Hypertension

The definition of pulmonary hypertension (PH), as established by the American Thoracic Society and American Heart Association (ATS/AHA) and the European Society of Cardiology and European Respiratory Society (ESC/ERS), is primarily applied to older paediatric patients. This definition does not account for the physiological decline in mean pulmonary artery pressure (mPAP) that occurs within the first three months of life. Careful analysis of large haemodynamic databases has challenged the empirical threshold for PH of 25 mmHg (velocity 2.5 m/s). Consequently, the 6th World Symposium on Pulmonary Hypertension (WSPH) in 2019 redefined PH as an mPAP ≥ 20 mmHg (velocity 2.24 m/s) and incorporated an indexed pulmonary vascular resistance (PVRi) threshold of ≥ 3 Wood Units × m^2^ (WU × m^2^) to specifically identify pre-capillary pulmonary hypertension [5,6,7].

### 1.2. Diagnostic Approach to PH in Extremely Premature Infants

Clinical feature of pulmonary hypertension (PH) in preterm infants is non-specific. Infants with PH may be asymptomatic or present with hypoxemic respiratory failure. Continuous monitoring of oxygen saturation and assessment of partial pressure of arterial carbon dioxide (PaCO_2_) on blood gas analysis as standard monitoring can aid in the recognition and evaluation of PH [8]. A preductal-to-postductal oxygen gradient—defined as a difference in arterial partial pressure of oxygen (PaO_2_) ≥ 20 mmHg or oxygen saturation (SpO_2_) ≥ 10%—is a significant indicator of PH in established cases. However, this gradient may be absent in infants with early-stage disease or in those with significant interatrial shunting [9].

The definitive diagnosis of PH requires measurement of the mean pulmonary arterial pressure (mPAP) via right heart catheterization. However, this invasive procedure is not feasible in critically ill preterm infants due to the associated risks of instrumentation, anaesthesia, and ionizing radiation exposure [10].

Echocardiography has emerged as the primary non-invasive screening tool for PH in preterm infants, particularly in the early neonatal period [11]. Echocardiographic findings highly suggestive of PH (in the absence of right ventricular outflow tract obstruction) include an elevated tricuspid regurgitant (TR) jet velocity ≥ 2.24 m/s (≥20 mmHg), a right-to-left shunt across the foramen ovale, and flattening of the interventricular septum [12]. Nevertheless, echocardiographic assessment has several limitations: it is operator’s technical skills-dependent and lacks standardized diagnostic definitions, so can be technically challenging. A reliable TR jet Doppler signal, crucial for pressure estimation, is obtainable in only approximately 61% of paediatric studies. Furthermore, when compared with cardiac catheterization, echocardiography demonstrated a sensitivity of 79% for diagnosing PH and accurately determined disease severity in only 47% of cases [13,14].

Despite these limitations, echocardiography remains the principal screening modality for the diagnosis of PH in premature infants.

### 1.3. Definition of Bronchopulmonary Dysplasia (BPD)

The diagnosis of BPD in preterm infants is traditionally defined as supplemental oxygen use either at a corrected age of 36 gestational weeks PMA or after 28 days of postnatal life in infants born <32 weeks of gestational age. The definition was revised in a 2018 workshop sponsored by the National Institute of Child Health and Human Development (NICHD) to incorporate modern non-invasive ventilation modalities and to provide a more detailed stratification of the fraction of inspired oxygen (FiO_2_) across various respiratory support strategies [15]. Subsequently, Jensen et al. (2019) proposed a further refined, evidence-based definition designed to improve the prediction of long-term respiratory and neurodevelopmental outcomes [16].

### 1.4. Bronchopulmonary Dysplasia-Associated Pulmonary Hypertension (BPD-PH)

BPD-PH develops primarily in survivors of extreme preterm birth as a consequence of incomplete pulmonary development, postnatal vascular remodelling triggered by hyperoxia and/or hypoxia, and arrested vascular growth. While the precise pathogenic mechanisms remain incompletely elucidated [12], this condition is associated with poor outcomes in extremely premature infants, including high rates of morbidity and mortality.

Current guidelines from the American Heart Association and American Thoracic Society (AHA/ATS) recommend screening for pulmonary hypertension in infants with established bronchopulmonary dysplasia at a corrected age of 36 gestational weeks PMA. However, emerging evidence suggests potential value in earlier echocardiographic assessment. Mourani et al. documented early pulmonary hypertension in up to 42% of preterm infants at seven days of age and proposed that its presence may represent a significant risk factor for subsequent development of BPD, prolonged oxygen dependency, and increased mortality [17].

The absence of consensus regarding standardized screening protocols and diagnostic criteria continues to obscure the true incidence of BPD-PH. A comprehensive understanding of its epidemiology, risk factors, and natural history is needed to establish evidence-based screening guidelines, develop effective preventive strategies, and optimize targeted therapeutic interventions [5].

## 2. Materials and Methods

The primary objectives of this study were: (i) to determine the incidence of pulmonary hypertension (PH) and bronchopulmonary dysplasia (BPD) in preterm infants of extremely low gestational age; (ii) to determine the incidence of PH among infants diagnosed with BPD (BPD-PH); and (iii) to evaluate the utility of early and late echocardiographic screening for PH; in infants with established BPD.

### 2.1. Study Participants

We conducted a prospective study involving preterm infants with a gestational age between 22 weeks, 0 days, and 28 weeks, 6 days, who were admitted to the Neonatal Intensive Care Unit (NICU) at the Maternity Hospital of the University Medical Centre Ljubljana, Slovenia, between 1 September 2022, and 31 December 2024. In this cohort, we screened for the presence of pulmonary hypertension (PH) and bronchopulmonary dysplasia (BPD).

The study was designed as a serial screening for PH, with assessments performed on day 7 of life (early screening) and at discharge.

BPD was diagnosed in infants who had supplemental oxygen dependency either at a corrected gestational age of 36 weeks p.m. or after 28 days of postnatal life in infants born at less than 32 weeks of gestation. BPD was graded according to NICHD 2018 consensus criteria [11].

The diagnosis of PH was confirmed using a combination of clinical and echocardiographic criteria, supported by a positive response to targeted pulmonary vasodilator therapy.

Clinically, PH was suspected in cases of hypoxic respiratory failure that was unresponsive to oxygen supplementation, mechanical respiratory support, or surfactant replacement therapy. Additionally, a preductal-to-postductal oxygen saturation (SpO_2_) difference of ≥10% was considered indicative. A significant diagnostic criterion was a positive clinical and oxygenation response to inhaled nitric oxide (iNO) or sildenafil treatment.

### 2.2. Echocardiographic Studies and Measurements

In our NICU, echocardiography is performed routinely within the first days of life for infants with severe respiratory distress syndrome requiring intensive care, repeated doses of surfactant, and/or inhaled nitric oxide due to clinical suspicion of pulmonary hypertension associated with RDS. In less severe cases, echocardiography is performed by day 7 as part of our screening for a patent ductus arteriosus.

Echocardiographic measurements were obtained using a 2022 GE HealthCare LOGIQ ultrasound system (GE HealthCare, Chicago, IL, USA) equipped with a GE 12S-RS neonatal cardiac transducer (5.0–11.0 MHz) (GE HealthCare, Chicago, IL, USA). In Slovenia, all pregnancies undergo a standard morphological screening program during foetal life, which includes evaluation for congenital heart defects. Despite this prenatal screening, each newborn in our study underwent a comprehensive postnatal echocardiographic assessment to definitively exclude congenital heart malformations and right ventricular outflow tract obstruction.

Diagnostic echocardiographic signs included a tricuspid regurgitant (TR) jet velocity greater than 2.24 m/s (20 mmHg), evidence of a right-to-left shunt across the foramen ovale and/or patent ductus arteriosus (PDA), and flattening of the interventricular septum. Right ventricular (RV) function was qualitatively assessed based on RV geometry, categorized as round (O-shaped), D-shaped, or crescent-shaped.

All echocardiographic examinations in this study were performed as a primary screening tool by senior neonatologists trained in neonatal echocardiography. The main goal of our study was to screen infants who, according to our current clinical practice, would not otherwise be referred to a cardiologist. Performing a complex echocardiographic examination—including advanced RV parameters such as TAPSE, strain, or tissue Doppler—was beyond the scope and outside the purpose of this screening.

In critically ill infants with a strong suspicion of pulmonary hypertension who required inhaled nitric oxide (iNO), if a senior neonatologist trained in echocardiography was available, echocardiography was performed at any time (outside the scheduled screening days) to assess pulmonary hypertension and to exclude cyanotic congenital heart disease. The unavailability of echocardiography was not a contraindication to starting iNO and did not justify delaying treatment.

For late screening, echocardiography was not a reason to retain infants in hospital until 36 + 0 weeks’ postmenstrual age; the screening echocardiogram was performed before discharge and before 36 + 0 weeks’ gestation.

## 3. Results

During the study period, 78 infants born between 22 + 0 and 28 + 6 weeks gestation were enrolled. Of these, 71 underwent early screening. Among the remaining seven infants, one was excluded due to congenital pulmonary valvular stenosis, one was transferred to the surgical ward immediately after birth, and five did not undergo echocardiography within the first 14 days due to technical or organizational reasons.

Of the 71 infants who underwent early screening, six later died, six were discharged home before 36 weeks postmenstrual age, one infant with pulmonary stenosis was identified. Retrospectively, we identified that in three infants, late screening was not performed because the senior neonatologists were not available at the time of discharge, and the attending colleagues inadvertently omitted them from the study. Consequently, late screening was performed in 57 infants. A flow chart of the infant cohort is shown in Figure 1.

The prenatal (maternal) characteristics, postnatal characteristics, morbidity, and mortality of the 71 infants who underwent early screening are presented in Table 1. For necrotising enterocolitis (NEC), we included infants with Stage II or higher, indicating definite NEC according to Bell’s criteria. For intraventricular haemorrhage (IVH), only severe Grade III/IV cases were included, while for retinopathy of prematurity (ROP), all grades were included. The diagnosis of ROP was made exclusively by an ophthalmologist, who screened all infants born before 29 weeks’ gestation on a weekly basis. For surfactant replacement, we primarily follow the European Consensus Guidelines on the Management of Respiratory Distress Syndrome (2022 update) [18]. Surfactant is administered early when FiO_2_ ≥ 0.30–0.40 is required to maintain adequate oxygenation on CPAP ≥6 cmH_2_O, or when there is clinical or radiographic evidence of worsening RDS. Non-invasive surfactant administration is preferred whenever the infant’s clinical condition allows. Notably, a high proportion of infants (46.15%) were treated with low-dose dexamethasone according to the DART protocol [19], prenatal corticosteroid administration at 74.6%, and postnatal surfactant replacement in 70.4% of infants in the cohort.

The median day of life for early screening was day 7 (mean 6.9 ± 2.5 days). On the day of screening, 13 infants (18.3%) required no ventilatory support or supplemental oxygen; 63.4% required supplemental oxygen, 39.4% required noninvasive ventilatory support, and 42.3% required invasive ventilatory support.

The median day of life for late screening was day 70 (mean 66.5 ± 15.2 days), corresponding to a median postmenstrual age of 35 weeks (mean 34.9 ± 1.5 weeks). On the day of late screening, 28 infants (49.1%) required supplemental oxygen, 7 (12.3%) required invasive respiratory support, and 7 (12.3%) required noninvasive respiratory support.

Echocardiographically, tricuspid regurgitation (TR) was absent in 18.3% of infants at early screening and in 68% at late screening. The mean TR jet velocity was 0.97 m/s (± 0.46) at early screening and 0.93 m/s (± 0.47) at late screening. A right-to-left shunt through a patent ductus arteriosus (PDA) was detected in 5 infants (7.0%) at early screening, and through a patent foramen ovale (PFO) in 10 infants (14.1%). No right-to-left shunts at the level of the PDA or PFO were detected at late screening. Echocardiographic characteristics at early and late screening are shown in Table 2 and Table 3, respectively.

Overall, pulmonary hypertension (PH) was diagnosed in 10 infants at early screening and in 1 infant at late screening.

All 10 infants diagnosed with early PH had severe respiratory distress that was unresponsive to oxygen therapy, surfactant replacement, and various ventilatory strategies. Each infant demonstrated a good clinical response to inhaled nitric oxide. Echocardiographic signs of PH (defined as a TR jet velocity >2.24 m/s and/or a right-to-left shunt through a PDA or PFO) were present in 6 of the 10 infants (60%).

The single patient diagnosed with PH in the late screening group was identified on day of life 96 (at a postmenstrual age of 38 weeks). Echocardiography demonstrated flattening of the interventricular septum.

The infant responded well to sildenafil therapy. The clinical and echocardiographic characteristics of infants with PH are summarized in Table 4.

Bronchopulmonary dysplasia (BPD) was diagnosed in 42 of the 57 infants (73.6%) in the late screening cohort. Notably, all infants who had pulmonary hypertension (PH) at the early screening and survived to the late screening developed BPD, all developed BPD grade I, classified according to the revised NICHD criteria. The distribution of BPD by gestational age is shown in Table 5.

## 4. Discussion

The aim of our study was to assess the incidence of pulmonary hypertension (PH) in extremely low gestational age neonates (ELGANs), its association with BPD, and outcomes at day 7 and at 36 weeks postmenstrual age (PMA), prior to discharge from the NICU.

PH associated with bronchopulmonary dysplasia (BPD-PH) is well established as a predictor of poor outcomes, including high morbidity and mortality [3]. However, within these time frames, we did not detect any infants with echocardiographic signs of PH. Only 10 infants with severe RDS, requiring the highest level of NICU support, had echocardiograms in the first days after admission (median day 2 of life). These revealed PH, and the infants required inhaled nitric oxide in addition to standard therapy. When echocardiography was repeated at the scheduled day 7 assessment, PH was no longer evident.

A meta-analysis of 44 observational studies including 7677 preterm infants reported pulmonary hypertension (PH) incidences of 5%, 18%, and 41% in mild, moderate, and severe BPD, respectively [20]. In recent European studies, an English cohort published in 2023 [21] reported a 13% incidence of late PH, while a Dutch study (2021) reported 5% [22]. Nonetheless, owing to the absence of consensus diagnostic criteria for PH in ELGANs with BPD, some experts argue that the true incidence of BPD-PH is unknown and likely underestimated [23]. Accordingly, the American Heart Association/American Thoracic Society (AHA/ATS) recommends screening for BPD-PH at 36 weeks PMA [5]. Other authors advocate even earlier screening; for example, Mourani et al. identified early PH in 42% of infants and linked it to increased BPD severity, prolonged oxygen dependence, and higher first-year mortality [17]. Despite the wide range of reported incidences of early pulmonary hypertension (PH) in recent studies—ranging from 8% to 55%—there is a notable trend toward decreasing incidence. For example, in the study by Mullaly et al. (2025) [24], early PH was identified in 20 of 166 infants (12%). This decline is understandable given the advances in the management of neonatal respiratory distress [9,24].

The main findings of our study are as follows: (i) The incidence of bronchopulmonary dysplasia (BPD) in the late-screening cohort was 73.6% which is in the range of recent published data registries as Australian and New Zealand Neonatal Network (ANZNN), Vermont Oxford Network (VON) and Canadian Neonatal Network (CNN) [25,26,27]. This rate is slightly higher than the most recent Swedish report (2025) [28], in which the incidence of any BPD was 65% and has remained unchanged in recent years. (ii) There were notably lower rates of both early and late pulmonary hypertension (PH) than those reported in the literature. Within the first 7 days of life, PH was diagnosed in 10 of 71 infants (13.9%), substantially below the 42% reported by Mourani et al. [17], and 26% in Dutch study 2021 [22] At late screening (median PMA 35 ± 1 weeks), PH was identified in only 1 of 57 infants (1.8%).

Three principal hypotheses may explain this discrepancy.

First, despite systematic screening, our methodology may have failed to capture all cases of PH. Echocardiograms were performed by trained neonatologists rather than paediatric cardiologists. Assessments were limited to tricuspid regurgitant (TR) jet velocity > 2.24 m/s, right-to-left shunting across the foramen ovale or PDA, and qualitative evaluation of RV morphology and septal position. However, reliable TR Doppler signals—crucial for pressure estimation—are obtainable in only about 61% of paediatric studies [13]. Furthermore, advanced functional RV assessments (e.g., TAPSE, FAC, tissue Doppler, 3D echo, or RV strain) were not performed, as these require subspecialty expertise and are not feasible in routine screening.

Supporting possible underdiagnosis, in all 11 identified PH cases diagnosis was primarily clinical—based on hypoxemic respiratory failure unresponsive to conventional therapy and/or positive response to iNO or sildenafil. Echocardiographic confirmation was achieved in 7 of cases (63.1%).

Research specifically addressing BPD-PH in ELGANs remains limited, indicating that this area is still under-investigated. Among recent studies, we identified two with designs broadly comparable to ours. However, both differ from ours in that they are retrospective. Branescu et al. [21] had echocardiograms reviewed retrospectively by a single pediatric cardiologist, whereas in the study by Arjaans et al. [22], the diagnosis of PH was made clinically, as in our study. The echocardiographic parameters considered included systolic pressure in the right ventricle (RVSP), the presence of a right-to-left shunt through a persistent ductus arteriosus, and flattening of the interventricular septum, without more advanced parameters such as TAPSE, FAC, or strain analysis. Interestingly, the observed incidence of late PH in the study by Arjaans et al. was lower (5%) than in that by Branescu et al. (13%), where echocardiograms were assessed by a cardiologist.

We did not identify any studies that systematically screened ELGANs for PH using echocardiography performed by non-cardiologists, which may represent a distinctive contribution of our work—particularly in settings where a paediatric cardiologist is not immediately available in the NICU. Accordingly, our primary aim was to screen infants who, under routine clinical practice, would not otherwise be referred to a cardiologist. We recognise, however, that this is also a key limitation: routine echocardiographic screening by neonatologists did not reliably enhance PH detection. These findings underscore the need to refine and standardise the screening approach in collaboration with paediatric cardiologists. If screening performed by neonatologists leads to under-diagnosis, regular cardiology involvement should be considered and/or structured training programmes for neonatologists implemented, with appropriate internal and external validation of the screening methodology.”

Second, the typical time for first detection of PH in the literature is 36–40 weeks PMA [29,30,31]. Current AHA/ATS guidelines therefore recommend screening infants with established BPD at 36 weeks PMA [5]. However, these recommendations are based on expert opinion or small studies. A systematic review showed that PH screening in preterm infants has varied widely, from 28 days of life to 36 weeks corrected GA, and even beyond 2 months of age [31].

In our study, the median GA at late screening was 35 ± 1 weeks. —meaning that most infants had not yet reached the guideline-recommended age of 36 weeks PMA. The main reason was early discharge, which, along with mortality, contributed to attrition between the early and late screening cohorts. As a result, screening before 36 weeks may have missed cases of PH developing later in the course of BPD. Thus, the timing of our late screening may have underestimated the true incidence of PH.

Third, the true incidence of PH in our cohort may have been low, potentially due to high-quality perinatal care. Prenatal corticosteroids, early surfactant administration, and caffeine therapy are known to reduce the risk of RDS, BPD, and associated complications. In our cohort, antenatal corticosteroids were given to 74.6% of mothers, surfactant to 70.5% of infants, and caffeine to all premature infants from day 1 of life until 34 weeks PMA.

In addition, 46.5% of infants received postnatal dexamethasone, which may reduce the risk of severe BPD and its complications, including PH. The use of postnatal corticosteroids for the prevention or treatment of BPD is common but variable: recent studies [32,33,34] and registry data from the Australian and New Zealand Neonatal Network (ANZNN) [26] and the Vermont Oxford Network (VON) [27,33] report rates ranging from 13% to 36%. In a recently published French multicentre study (2023) [31] involving 13,913 infants (mean birth weight 1145 ± 366 g), 26.1% were exposed to postnatal corticosteroids, including 21.8% via systemic and 10.1% via inhaled routes.

Despite a rising trend (e.g., from 13% in 2013 to 22% in 2021 per ANZNN), the 46.2% rate in our cohort is comparatively high. Although postnatal dexamethasone can impair pulmonary angiogenesis and potentially increase pulmonary vascular resistance [35] it reduces mortality and the incidence and severity of BPD [36] Since PH risk rises with BPD severity—6% in mild, 12% in moderate, and 39% in severe cases [36]—a lower rate of severe BPD would be expected to correspond with reduced BPD-PH.

The third hypothesis is based purely on assumptions, which can be clarified only after excluding the possibility of underdiagnosis of PH by neonatologists, as mentioned above.

Finally, regarding the incidence of bronchopulmonary dysplasia-associated pulmonary hypertension (BPD-PH)—a predefined objective of our study—BPD was diagnosed in 42 of 57 infants (73.7%). The distribution across gestational ages is shown in Table 5 and is consistent with recent studies and registry data: in contemporary cohorts, BPD is extremely common at the lowest gestations—approximately 90–93% at 23–24 weeks, 75–78% at 25–26 weeks, and ~54% at 27–28 weeks [26,27,28,37].

Among these 42 infants with BPD, pulmonary hypertension was identified in only one case (~2.4%). As noted earlier, a meta-analysis of 44 observational studies including 7677 preterm infants reported PH incidences of 5%, 18%, and 41% in mild, moderate, and severe BPD, respectively. This suggests a lower BPD-PH incidence in our cohort than reported elsewhere [20,38]; however, given the single observed case, the true incidence among extremely low-gestational-age infants in our study cannot be estimated with precision.

In summary, routine echocardiographic screening performed by neonatologists in our study did not reliably improve detection of PH at either 7 days of life or at median 35 weeks PMA. These findings present an important challenge. To improve detection, we must both strengthen expertise in echocardiographic interpretation and consider extending screening beyond the neonatal period, as PH may not manifest until after 36 weeks PMA. Addressing this will require collaborative refinement of screening protocols with paediatric cardiologists, incorporating standardized, feasible, and sensitive echocardiographic parameters for PH detection in this vulnerable population.

## 5. Conclusions

In this single-centre cohort study, routine echocardiographic screening for pulmonary hypertension (PH) performed by neonatologists did not improve detection rates in extremely low gestational age neonates.

This finding may be attributed to (i) limitations in the screening methodology itself—specifically, performing echocardiographic screening by neonatologists rather than paediatric cardiologists—which may have led to an underestimation of the true incidence; (ii) the early timing of screening, prior to 36 weeks postmenstrual age (PMA); or (iii) a genuinely low incidence of PH, potentially resulting from high-quality perinatal care, including the widespread use of postnatal corticosteroids. However, this remains an assumption that requires further investigation.

Taken together, the results underscore a significant clinical challenge in diagnosing PH in this vulnerable population. To improve detection, a collaborative review of the current screening protocol with paediatric cardiologists is essential to establish more sensitive and standardized echocardiographic criteria. Furthermore, consideration should be given to extending screening beyond the neonatal period and hospital discharge, as PH may not manifest until after 36 weeks PMA.

## Figures and Tables

**Figure 1 children-12-01441-f001:**
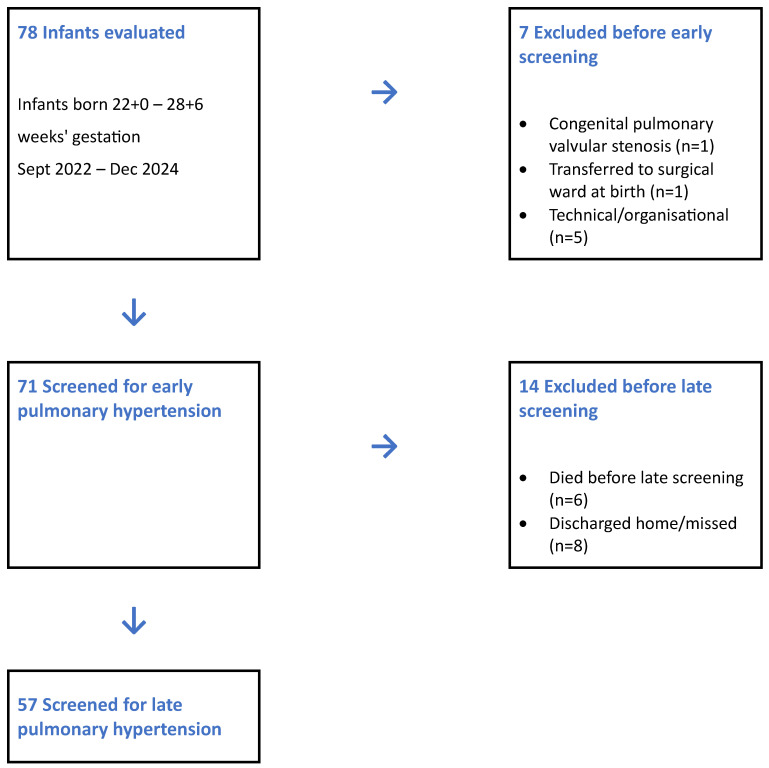
Flow chart of the patients evaluated in the study.

**Table 1 children-12-01441-t001:** Maternal and neonatal characteristics in the PH screening cohort.

	Total (N = 71)
Neonatal	
Characteristic	
Gestational age, weeks—median (IQR)	25 (24–27)
Sex—Male, n (%)	40 (56.3%)
Sex—Female, n (%)	31 (43.7%)
Small for gestational age (SGA), n (%)	14 (19.7%)
Apgar score at 1 min—median (IQR)	6 (4–7)
Apgar score at 5 min—median (IQR)	7 (6–8)
Postnatal steroids, n (%)	30 (46.15%)
Surfactant, n (%)	50 (70.4%)
Caffeine, n (%)	71 (100%)
Morbidity and mortality	
Mortality, n (%)	6 (8.5%)
PDA present, n (%)	38 (53.5%)
PVL, n (%)	5 (7.0%)
NEC, n (%)	5 (7.0%)
ROP, n (%)	33 (46.5%)
IVH grade 3/4, n (%)	7 (9.9%)
BPD, n (%)	49 (69.0%)
Maternal	
Cesarean section, n (%)	28 (39.4%)
Prenatal steroids, n (%)	53 (74.6%)
Maternal age >37 years, n (%)	12 (16.9%)
Preeclampsia, n (%)	4 (5.6%)
Eclampsia, n (%)	0 (0.0%)
Gestational diabetes, n (%)	5 (7.0%)

Abbreviations: PDA = patent ductus arteriosus; PVL = periventricular leukomalacia; NEC = necrotising enterocolitis; ROP = retinopathy of prematurity; IVH = intraventricular hemorrhage; BPD = bronchopulmonary dysplasia.

**Table 2 children-12-01441-t002:** Early PH screening: clinical and echocardiographic features.

	Total N = 71
Characteristic	
Day of early screening (median [IQR]; mean ± SD), days	7 (7–7); 6.9 ± 2.5
Oxygen need at early screening, n (%)	45 (63.4%)
Noninvasive ventilation (NIV) at early screening, n (%)	28 (39.4%)
Invasive ventilation (IMV) at early screening, n (%)	30 (42.3%)
No ventilatory support (neither NIV nor IMV) at early screening, n (%)	13 (18.3%)
Echocardiography	
No tricuspid regurgitation (TR = N), n (%)	12 (16.9%)
TR jet velocity (m/s), mean ± SD	0.97 ± 0.46
PDA present (any L→R or R→L), n (%)	38 (53.5%)
R→L shunt in PDA: n (%) of total cohort; % of PDA cases	5 (7.0%); 13.2%
PFO present (any L→R or R→L), n (%)	64 (90.1%)
R→L shunt in PFO: n (%) of total cohort; % of PFO cases	10 (14.1%); 15.6%
RV dilatation, IV septum flattened, n (%)	0 (0%)

Abbreviations: TR = tricuspid regurgitation; PDA = patent ductus arteriosus; PFO = patent foramen ovale; L→R = left-to-right shunt; R→L = right-to-left shunt; RV = right ventricle; IV = interventricular.

**Table 3 children-12-01441-t003:** Late PH screening: clinical and echocardiographic features.

	Total N = 57
Characteristic	
Day of late screening (median [IQR]; mean ± SD), days	70 (56–77); 66.5 ± 15.2
Oxygen need at late screening, n (%)	28 (49.1%)
Noninvasive ventilation (NIV) at late screening, n (%)	7 (12.3%)
Invasive ventilation (IMV) at late screening, n (%)	7 (12.3%)
No ventilatory support at late screening, n (%)	43 (75.4%)
Echocardiography	
No tricuspid regurgitation (TR = N), n (%)	36 (63.2%)
TR jet velocity (m/s), mean ± SD	0.93 ± 0.47
PDA present (any L→R or R→L), n (%)	10 (17.5%)
R→L shunt in PDA: n (%) of total cohort; % of PDA cases	0 (0.0%); 0.0%
PFO present (any L→R or R→L), n (%)	20 (35.1%)
R→L shunt in PFO: n (%) of total cohort; % of PFO cases	0 (0.0%); 0.0%
RV dilatation, IV septum flattened, n (%)	1 (1.75%)
Gestational age at late screening (median [IQR]; mean ± SD), weeks	35 (34–36); 34.9 ± 1.5

Abbreviations: TR = tricuspid regurgitation; PDA = patent ductus arteriosus; PFO = patent foramen ovale; L→R = left-to-right shunt; R→L = right-to-left shunt; RV = right ventricle; IV = interventricular.

**Table 4 children-12-01441-t004:** Characteristics of infants with pulmonary hypertension.

Patient ID	GA (Weeks)	Day of Diagnosis	Survival	Diagnosis of PH						BPD
				Clinical		Echocardiography				
				FiO_2_	iNO/Sildenafil Response	TR Jet (m/s)	PDA	PFO	RV Dilatation/Septum Flattering	
Early screening										
1	26	4	S	1.0	+	0.85	−	−	−	+
2	24	1	S	0.6	+	−	−	−	−	+
3	26	5	S	1.0	+	0.80	−	−	−	+
4	24	1	S	1.0	+	0.72	L→D	L→D	−	+
5	22	1	D	1.0	+	3.06	D→L	D→L	−	/
6	26	2	S	1.0	+	0.90	D→L	D→L	−	+
7	25	2	S	1.0	+	−	D→L	D→L	−	+
8	27	2	S	0.7	+	1.20	D→L	D→L	−	+
9	25	2	D	0.6	+	0.60	L→D	D→L	−	/
10	23	1	D	1.0	+	3.00	D→L	D→L	−	/
Median (IQR)	25 (24–26)	2 (1–2)	/	1.0 (0.77–1.0)	/	0.88 (0.78–1.65)	/	/	/	/
Late screening										
11	24	96	S	0.4	+	−	L→D	L→D	+	+

Abbreviations: “+” = present; “−” = absent; GA = gestational age (weeks); D = died; S = survivor; BPD = bronchopulmonary dysplasia; PH = pulmonary hypertension; PDA = patent ductus arteriosus; PFO = patent foramen ovale; L→D = left-to-right shunt; D→L = right-to-left shunt; TR jet = tricuspid regurgitation Doppler jet velocity; “/” = not applicable/not existent.

**Table 5 children-12-01441-t005:** BPD across gestational ages in the late screening cohort.

GA (Weeks)	All n = 57 (%)	BPD n = 42 (%)	% with BPD
23	3 (5.3)	3 (7.1)	100.0
24	14 (24.6)	12 (28.6)	85.7
25	13 (22.8)	12 (28.6)	92.3
26	8 (14.0)	6 (14.3)	75.0
27	11 (19.3)	6 (14.3)	54.5
28	8 (14.0)	3 (7.1)	37.5

Abbreviations: GA = gestational age; BPD = bronchopulmonary dysplasia.

## Data Availability

The data supporting the findings of this study are available from the corresponding author upon reasonable request.

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
