# Peer review of "The Incidence of Pulmonary Hypertension and the Association with Bronchopulmonary Dysplasia in Preterm Infants of Extremely Low Gestational Age: Single Centre Study at the Maternity Hospital of University Medical Centre Ljubljana, Slovenia"

_children, 2025, doi:10.3390/children12111441_

Round 1
Reviewer 1 Report
Comments and Suggestions for Authors
Dear Colleagues,
I had to review your original article entitled ”The incidence of pulmonary hypertension and the association with bronchopulmonary dysplasia in preterm infants of extremely low gestational age: single centre study at the Maternity Hospital of University Medical Centre Ljubljana, Slovenia”, a paper covering an interesting aspect of bronchopulmonary dysplasia (BPD), namely the associated pulmonary hypertension (HP). Here are my observations:
- The title was well chosen, reflecting the content of the research.
- Well-pointed highlights precede the abstract.
- The well-structured abstract clearly presents the most important points of the research.
- The keywords were correctly chosen.
- The Introduction section offers sufficient information on the discussed topic, BPD and PH. The study's aims are clearly defined.
- In the Materials and Methods section, the authors could provide definitions for necrotizing enterocolitis, intraventricular hemorrhage, and retinopathy of prematurity, specifying whether cases of all severities were included in the analysis. A good addition is the criteria used to administer surfactant in this category of population, as only 2/3 of the preterm infants included in the study received surfactant. This is also important to consider when the authors discuss the study's limitations.
- Regarding the results, please offer more information on the patient diagnosed at 38 weeks postmenstrual age with PH associated with BPD (gestational age, birth weight, IUGR status, RDS severity, for example). Was this patient already discharged?
- Additionally, please consider classifying patients diagnosed with PH according to both the BPD severity and the PH classification. Thus, comparisons can be made with other data in the literature.
- Dexamethasone administration postnatally may have influenced the BPD severity, it is true, yet, as stated in line 311, ”postnatal dexamethasone can impair pulmonary angiogenesis and potentially increase pulmonary vascular resistance (28)”; please, consider offering more details.
- When discussing the limitations, indeed, PH incidence is lower compared to the data offered by Mourani et al. However, that study was published in 2008, and significant advances have been made in RDS management since that time. Please consider finding more recent data for comparison or reconsider the statement.
- Tables are well presented and designed.
- Figure 1 is difficult to read.
- The conclusions align with the study's results.
Congratulations.
Author Response
Response to Reviewer 1
Dear colleague,
Thank you for your review. We appreciate your positive evaluation and your thoughtful, professionally grounded comments, to which we have provided detailed responses.
1) Definitions for NEC, IVH, and ROP; criteria for surfactant administration
Thank you for pointing out the missing definitions. For necrotising enterocolitis (NEC), we included infants with Stage II or higher, indicating definite NEC according to Bell’s criteria. For intraventricular haemorrhage (IVH), only severe Grade III/IV cases were included, while for retinopathy of prematurity (ROP), all grades were included. The diagnosis of ROP was made exclusively by an ophthalmologist, who screened all infants born before 29 weeks’ gestation on a weekly basis.
For surfactant replacement in infants <29 weeks’ gestation, we primarily follow the European Consensus Guidelines on the Management of Respiratory Distress Syndrome (2023 update). Surfactant is administered early when FiOâ‚‚ ≥0.30–0.40 is required to maintain adequate oxygenation on CPAP ≥6 cmHâ‚‚O, or when there is clinical or radiographic evidence of worsening RDS. Non-invasive surfactant administration is preferred whenever the infant’s clinical condition allows.
The missing definitions and criteria for surfactant replacement have been added to the revised text.
2) Detailed information for the patient diagnosed at 38 weeks PMA
The comment regarding additional details on the patient diagnosed with pulmonary hypertension (PH) at 38 weeks PMA is valid. The patient is a female born at 24 weeks’ gestation, birth weight 560 g (just above the 10th percentile). She was intubated in the delivery room, received surfactant immediately after admission and again at 24 hours, and was mechanically ventilated until day 46. Dexamethasone was administered twice—on day 43 and day 95—both according to the DART protocol. On day 96, echocardiography confirmed PH (mild right heart dilation, flattened septum), and sildenafil was initiated. After the first DART course, she was extubated; after the second DART and sildenafil, noninvasive ventilation was briefly replaced by oxygen cannulas, but this was unsuccessful, requiring a return to noninvasive support. At day 105 (40 weeks PMA), she was transferred to the Pediatric Clinic for further management. The missing data have been included in the text as suggested.
3) Classification of BPD in patients with BPD
Thank you for this important observation. Of the seven survivors with early PH, all developed BPD grade I, classified according to the revised NICHD criteria. This information has been added to the Results section.
4) Dexamethason administration postnatally
While finalising the results of our study and exploring possible explanations, we formulated three hypotheses. After clarifying and emphasising the limitations of echocardiographic screening performed solely by neonatologists, and considering the timing of late screening, we reflected on potential reasons why the incidence of PH in ELGANs might truly be low. We therefore reviewed the available maternal and neonatal characteristics of our cohort, among which postnatal steroid use was the only factor that stood out.
We are fully aware that postnatal dexamethasone may impair pulmonary angiogenesis and consequently increase pulmonary vascular resistance. However, it also clearly reduces mortality as well as the incidence and severity of BPD, which could in turn influence the incidence of BPD-associated PH. Of course, this remains an assumption that requires further investigation. We appreciate your valuable remark and have now clarified this point more explicitly in the text, adjusting the tone of the Discussion section accordingly.
5) Incidence of early PH
We appreciate your insightful comment regarding the changing incidence of early pulmonary hypertension (PH) with advances in neonatal management. This aspect had not been our primary focus in the initial version. Following your suggestion, we reviewed recent studies reporting lower incidences and have appropriately incorporated these data into the discussion, with corresponding adjustments to the tone of the text.
In the revised manuscript changes are marked in blue.
We thank the reviewer once again for their professional and well-founded comments, which we have endeavored to address. We hope that our responses are clear and satisfactorily answer the questions raised.
Reviewer 2 Report
Comments and Suggestions for Authors
Thank you for the opportunity to review this manuscript. Few suggestions to help improve the manuscript.
- The median PMA at late screening was 35 weeks, which is before the guideline-recommended 36 weeks PMA. Please clarify why late screening was performed earlier than recommended? Please discuss how this may have underestimated true PH incidence. A few infants were screened well before 36 weeks (Line 211); this needs clearer justification or rephrasing as "pre-discharge" screening rather than "late" screening. (Lines 189–190, 322–327)
- All echocardiographic assessments were performed by neonatologists, but this is not sufficiently discussed as a limitation in the main results or discussion. Please elaborate on how the lack of paediatric cardiologist involvement and the absence of advanced RV functional parameters (e.g. TAPSE, strain, tissue Doppler) may have contributed to underdiagnosis. Please also provide any internal validation process or inter-observer agreement, if available. (Lines 180–182, 328–335)
- Please clarify your exclusion criteria, specifically, were infants with severe congenital anomalies (beyond pulmonary stenosis) excluded? Also, please specify what “technical or organizational reasons” prevented echocardiograms in some cases (Line 196). Were these missing at random, or could they introduce selection bias? (Lines 140–146, 193–197)
- You report that several PH diagnoses were based primarily on clinical presentation and iNO response rather than definitive echocardiographic signs (Lines 225–227). This should be acknowledged more clearly as a diagnostic limitation and potential bias. Please clarify the thresholds used for diagnosis in these cases. Was this approach pre-specified, or clinical judgment-based? (Lines 222–231, 336–340)
- The explanation for low PH incidence largely leans on assumptions of high-quality care and steroid use. While plausible, please balance this more critically with the possibility of underdiagnosis due to early screening and limited imaging parameters. Consider revising the tone in lines 298–304 and 336–340 to acknowledge both sides with balanced perspectives.
- The manuscript aims to assess PH incidence and outcomes at “day 7” and “36 weeks PMA,” but the actual late screening was performed at a range of 34–36 weeks (Line 322). Please standardize terminology throughout (e.g., call it “pre-discharge screening” if not strictly at 36 weeks) and make it clear that outcome assessment timing was variable due to early discharges. (Lines 134–139, 270–272)
Author Response
Response to Reviewer 2
Dear reviewer,
Thank you very much for your thorough and insightful review. Please find below our response to your valuable comments.
1) The median PMA at late screening
The median PMA of 35 weeks for the late screening is indeed accurate in our cohort. This aligns with our clinical practice, where the treatment for extremely low gestational age newborns (ELGANs) is often completed before 36 weeks. Specifically, for infants without serious health complications who are preparing for discharge between 34 and 36 weeks, we perform the screening echocardiography prior to them leaving the hospital.
To avoid any potential confusion regarding the term late screening, which according to the guidelines refers to 36 weeks, we have revised the manuscript to consistently use the term »pre-discharge screening«, as you suggested.
2) Limitation of echocardiographic assessments performed by neonatologists
Our NICU is located in the Department of Perinatology (Maternity hospital) University Medical Centre Ljubljana, Ljubljana, Slovenia, which takes care of all mostly very preterm and extremely preterm inborn infants who need neonatal intensive care. University Children's Hospitalwhich is located in the other building has also Department of Neonatology, but primarly taking care of the infants transferred from the other Slovenian Maternity hospitals for various clinical problems.Department of Pediatric Cardiology is part of the University Children's Hospita and provides 24-hour cardiology service. Therefore, some senior neonatologists, through clinical practice and additional training, are qualified to perform basic echocardiographic assessments. When a serious cardiac condition is suspected, a pediatric cardiologist from the Department of Pediatric Cardiology either visits our unit or the infant is transferred to their department.
The main goal of our study was to screen infants who, according to our current clinical practice, werre never or would not otherwise be referred to a cardiologist. Since most neonatologists are not trained cardiologists, performing a complex echocardiographic examination — including advanced RV parameters such as TAPSE, strain, or tissue Doppler — is beyond their scope and outside the purpose of screening.
We have therefore emphasized this limitation in the revised Discussion section and adjusted the tone of the text to make this point clearer.
3) Exclusion criteria
During the study period, only one infant with pulmonary stenosis was identified. Apart from this case, no infants with severe congenital anomalies were found in our cohort.
Echocardiographic screening in this study was performed exclusively by two senior neonatologists. Retrospectively, we identified that in three infants, without any particular reason, late screening was not performed because the senior neonatologists were not available at the time of discharge, or the attending colleagues inadvertently omitted them from the study. A retrospective review confirmed that these three infants were healthy, without any respiratory or cardiac issues at discharge.
We have acknowledged this limitation in the revised manuscript and clarified it in the Methods section to ensure full transparency regarding the study sample.
4) PH diagnosis based on clinical presentation and iNO response
The research was conducted in parallel with routine clinical work, and we did not wish to alter clinical decision-making unless absolutely necessary. Therefore, in the described cases of PH, when the clinician decided to initiate iNO therapy due to a severe clinical presentation, it was later found that in 4 out of 10 infants, echocardiographic signs of PH were not detected during our screening. Nevertheless, these infants were treated with iNO based on clinical judgment.
This observation further supports the limitation of echocardiographic assessments performed by neonatologists, as discussed in Comment 2.
5) Explanation for low PH incidence based on assumptions of high-quality care and steroid use
As partly explained in our previous comments, we acknowledge that this remains an assumption. As suggested by the reviewer, we have therefore balanced the hypothesis in the text and adjusted the tone accordingly to reflect this more cautiously.
6) Standardization of terminology
As agreed in Comment 1, the terminology has been standardized throughout the text to “pre-discharge screening.”
In the revised manuscript changes are marked in red.
The authors would like to once again thank the reviewer for their thorough review and valuable comments. We hope that our revisions and explanations adequately address the reviewer’s well-founded observations.
Reviewer 3 Report
Comments and Suggestions for Authors
As a neonatologist who is daily confronted with critically ill preterm infants, I read your paper on the incidence of pulmonary hypertension and its association with bronchopulmonary dysplasia in the population of extremely immature neonates with great interest. The topic is of great importance, especially since pulmonary hypertension remains one of the key predictors of poor outcomes in this population.
Suggestions:
- Late screening was on average performed at 35 weeks PMA, which is earlier than the officially recommended 36 weeks. This significantly reduces the possibility of detecting PH later in the course of the disease. The authors themselves acknowledge this methodological limitation, but in the reviewed manuscript the impression remains that the conclusion about the low incidence of PH may be premature.
- Echocardiographic assessment was entirely conducted by neonatologists. While I appreciate the effort to make diagnostics available in everyday practice, the lack of involvement of pediatric cardiologists and the absence of more advanced parameters (TAPSE, FAC, strain analysis) represent a significant methodological weakness and may lead to underestimation of the true incidence of PH.
- A portion of the subjects were not included in the late screening due to mortality or early discharge. This raises the issue of selection bias, since it is precisely the most vulnerable infants, in whom the highest incidence of PH would be expected, that were not included in the final analysis.
- The authors state that the high rate of postnatal corticosteroid use may have contributed to the lower incidence of PH. However, it is known that corticosteroids may also have a potentially adverse impact on pulmonary vascular development. This dilemma deserves more detailed discussion, as the reader is left with a simplified impression of a protective effect.
- Although relevant references were used, a more up-to-date comparison with the most recent multicenter data (particularly from 2022-2024) is lacking. In the discussion it would be useful to refer to trends from larger registries, which would further emphasize the value of your center in the European context.
Author Response
Response to Reviewer 3
Dear Reviewer,
Thank you for your thoughtful and insightful comments. We have carefully considered your feedback, and our responses are provided below.
1) The median PMA at late screening
We agree with your observation that the timing of late screening can substantially reduce the likelihood of detecting PH later in the disease course. This is consistent with our clinical practice, where management of extremely low gestational age newborns (ELGANs) is usually completed before 36 weeks. In particular, for infants without major health complications who are approaching discharge between 34 and 36 weeks, echocardiographic screening is performed prior to hospital discharge.
To prevent any ambiguity surrounding the term late screening—which, according to the guidelines, refers to 36 weeks—we have revised the manuscript to consistently use the term pre-discharge screening.
2) Limitation of echocardiographic assessments to neonatologists
Our NICU is part of the Department of Perinatology at the University Medical Centre Ljubljana (Maternity Hospital), Slovenia. It provides care primarily for very preterm and extremely preterm inborn infants who require intensive neonatal support. The University Children’s Hospital, located in a separate building, also has a Department of Neonatology that mainly treats infants transferred from other Slovenian maternity hospitals for various clinical conditions. The Department of Paediatric Cardiology, which is part of the University Children’s Hospital, provides 24-hour cardiology services.
In our unit, several senior neonatologists have acquired the necessary skills to perform basic echocardiographic assessments through extensive clinical experience and additional training. When a serious cardiac abnormality is suspected, a paediatric cardiologist from the Department of Paediatric Cardiology either performs a bedside evaluation in our NICU or the infant is transferred for further assessment.
The primary objective of our study was to screen infants who, according to our current clinical practice, would not routinely be referred to a cardiologist. As most neonatologists are not trained cardiologists, advanced echocardiographic evaluation—including detailed right ventricular measurements such as TAPSE, strain analysis, or tissue Doppler imaging—was beyond the intended scope of this screening.
We have accordingly clarified this limitation in the revised Discussion section and adjusted the text to make this point more explicit.
3) Subjects excluded from late screening
All echocardiographic screenings were conducted by two senior neonatologists. Upon retrospective review, we found that late screening was inadvertently not performed in a few infants. This occurred either because the senior neonatologists were unavailable at the time of discharge or because the infants were unintentionally omitted by the attending staff. Retrospectively, these infants had no serious medical conditions, particularly no respiratory or cardiocirculatory issues. We acknowledge that this could introduce bias; therefore, we have addressed it as a limitation in the revised manuscript and clarified the description in the Methods section to ensure transparency regarding the study population.
4) High rate of postnatal steroid use
While finalising the results of our study and exploring possible explanations, we formulated three hypotheses. After clarifying and emphasising the limitations of echocardiographic screening performed solely by neonatologists, and considering the timing of late screening, we reflected on potential reasons why the incidence of PH in ELGANs might truly be low. We therefore reviewed the available maternal and neonatal characteristics of our cohort, among which postnatal steroid use was the only factor that stood out.
We are fully aware that postnatal dexamethasone may impair pulmonary angiogenesis and consequently increase pulmonary vascular resistance. However, it also clearly reduces mortality as well as the incidence and severity of BPD, which could in turn influence the incidence of BPD-associated PH. Of course, this remains an assumption that requires further investigation. We appreciate your valuable remark and have now clarified this point more explicitly in the text, adjusting the tone of the Discussion section accordingly.
6) Up-to-date references for the most recent multicentre data
Thank you for noting the absence of the most recent references, particularly in the European context. After reviewing primarily European literature, we identified a 2023 French multicentre study on postnatal steroid use and a recently published Swedish study on BPD; both are discussed in the Discussion and cited in the references.
Notably, research specifically addressing BPD-PH in ELGANs remains limited, indicating that this area is still under-investigated. We found two studies comparable to ours—a 2023 study from St George’s University Hospital, London, and a 2021 Dutch study—both retrospective. These are commented on in the Discussion and cited accordingly.
In the revised manuscript changes are marked in green.
We thank the reviewer once again for their professional and well-founded comments, which we have endeavored to address. We hope that our responses are clear and satisfactorily answer the questions raised.
Round 2
Reviewer 3 Report
Comments and Suggestions for Authors
Thank you to the authors for the additional clarifications and the revisions made.